# Comparative Analysis of Chemical Constituents in Different Parts of Lotus by UPLC and QToF-MS

**DOI:** 10.3390/molecules26071855

**Published:** 2021-03-25

**Authors:** Haotian Pei, Wenyu Su, Meng Gui, Mingjie Dou, Yingxin Zhang, Cuizhu Wang, Dan Lu

**Affiliations:** School of Pharmaceutical Sciences, Jilin University, Fujin Road 1266, Changchun 130021, China; peiht18@mails.jlu.edu.cn (H.P.); suwy19@mails.jlu.edu.cn (W.S.); guimeng19@mails.jlu.edu.cn (M.G.); doumj20@mails.jlu.edu.cn (M.D.); yingxin20@mails.jlu.edu.cn (Y.Z.); wangcuizhu@jlu.edu.cn (C.W.)

**Keywords:** lotus, UPLC-QToF-MS, phytochemicals, analysis, different parts, metabolomics

## Abstract

Six parts of lotus (seeds, leaves, plumule, stamens, receptacles and rhizome nodes) are herbal medicines that are listed in the Chinese Pharmacopoeia. Their indications and functions have been confirmed by a long history of clinical practice. To fully understand the material basis of clinical applications, UPLC-QToF-MS combined with the UNIFI platform and multivariate statistical analysis was used in this study. As a result, a total of 171 compounds were detected and characterized from the six parts, and 23 robust biomarkers were discovered. The method can be used as a standard protocol for the direct identification and prediction of the six parts of lotus. Meanwhile, these discoveries are valuable for improving the quality control method of herbal medicines. Most importantly, this was the first time that alkaloids were detected in the stamen, and terpenoids were detected in the cored seed. The stamen is a noteworthy part because it contains the greatest diversity of flavonoids and terpenoids, but research on the stamen is rather limited.

## 1. Introduction

*Nelumbo nucifera* Gaerth., an aquatic plant in the Nymphaeaceae family, is distributed in wetlands throughout Asia. It is also known as sacred lotus; it is not only used as a food and herb, but also deeply related to Buddhism in Asia, and its flower is used as the pedestal for divine figures. The medicinal functions of lotus were recognized earlier than its edible value and were recorded for the first time in the book “Er ya” (400 B.C.) [1]. Almost all parts of this plant have been used as food as well as herbal medicine for over 2000 years, and especially the lotus seed and rhizome are more widely used due to their delicious taste and great nutritive value [2]. The seed (Nelumbinis Semen), leaf (Nelumbinis Folium), plumule (Nelumbinis Plumula), stamen (Nelumbinis Stamen), receptacle (Nelumbinis Receptaculum) and rhizome node (Nelumbinis Rhizomatis Nodus) are listed in the official Pharmacopoeia of China (CP). Their indications and functions, which were confirmed by clinical practice for thousands of years, were recorded in the Chinese Pharmacopoeia. According to the records, lotus leaves, receptacles and rhizome nodes have a hemostatic effect; stamens can prohibit pathological spermatorrhoea and frequent urination; and the seeds and plumule can mind-tranquilize and improve sleep [3] (Figure 1).

In recent decades, increasing research has focused on this special herb, especially the plumule and leaf, and many of its constituents have been found to possess extensive features of health benefits. The ingredients and activities of the different lotus parts, together with their applications in the food and healthcare area, have similarities and differences [2]. Without a doubt, the pharmacological effects of any plant or effective part are based on their phytochemicals. Flavonoids that have been found in the six parts of lotus are associated with a variety effects, including antioxidant [4,5,6,7], anti-inflammatory [8], antiviral [9,10], anti-obesity [11,12], and antimicrobial effects [13]. Alkaloids are effective ingredients used for treating cardiovascular diseases [14,15,16], regulating blood lipids [17], tranquilizing the mind [18] and treating cancer [19]. However, the chemical bases of their activities are far from clear compared with the medical history of thousands of years. For example, stamens and receptacles are reported to have anti-ischemic effects [20]; meanwhile, neferine and liensinine were confirmed to be anti-ischemic agents [21], but there is no proof that stamens contain alkaloids. Thus, just what do the stamens and receptacles contain that contributes to the anti-ischemic effect? Furthermore, the material basis for traditional uses as hemostatic agents is a “blank” area. Moreover, there are no index components for the quality control of seeds, stamens, receptacles and rhizome nodes in CP. Hence, this study focuses on the chemical constituents of the six parts of lotus. 

UPLC is the best chromatographic method in terms of resolution, sensitivity, and speed. QToF-MS is the most sensitive quantitative and most comprehensive qualitative detector to identify and quantify the broadest range of compounds in the most complex and challenging samples. With its combined ability of high resolution and sensitivity, UPLC-QToF-MS has been successfully used for the analyses of complex samples [22]. It can be used for the rapid differentiation of different parts of a plant [23], identification of the habitats of herbs [24], and evaluation of the quality of TCMs and processed products [25]. The UNIFI information system has the ability to incorporate scientific library into with UPLC and QToF-MS data, which streamlines the process of identifying chemical structures in complex natural products. To obtain the chemical constituent profile of the cored seeds (**Sem**), leaves(**Fol**), plumule (**Plu**), stamens (**Sta**), receptacles (**Rec**) and rhizome nodes (**RN**) of lotus, we proposed a multiple ingredients identification strategy based on UPLC-QToF-MS coupled with the UNIFI informatics platform. This method can quickly identify multiple components. In this study, the constituents of the six parts were efficiently separated by ultra-performance liquid chromatography (UPLC) and detected by quadrupole time-of-flight tandem mass spectrometry (QToF-MS). Then, the data obtained by UPLC and QToF-MS were processed by the integrated information platform UNIFI.

As a result, a total of 171 components were identified from the six parts. Moreover, the differentiating components were screened by principal component analysis (PCA) and orthogonal projections to latent structures discriminant analysis (OPLS-DA). These methods provide a holistic and intuitionistic description of the chemical constituents in the six parts. Twenty-three robust biomarkers were found to distinguish the six parts. The established method can be used as a standard protocol for directly discriminating between and predicting the six parts of lotus. Most importantly, data analysis provided useful information for further study and usage of the plant.

## 2. Results

### 2.1. Identification of Components

A total of 171 compounds were identified or tentatively characterized from the six parts of lotus, including alkaloids, flavonoids, terpenoids, steroids, organic acids, etc. Among them, 84 compounds were identified in positive mode, and 127 compounds were identified in negative mode. The base peak intensity (BPI) chromatograms marked with the number of compounds are shown in Figure 2. The compound identification data were listed in Table 1. The chemical structures of the compounds are shown in Figure 3. More specifically, 86, 56, 89, 87, 60 and 19 compounds were identified from **Sem, Fol, Plu, Sta, Rec** and **RN**, respectively. By comparing the numbers of the detected compounds and signal strength of the chromatographic peak observed in the UPLC-QToF-MS assay, it seems that ESI^−^ mode is better than ESI^+^ for this test. However, running the ESI^+^ mode is still necessary because some compounds display better responses in ESI^+^ mode than in ESI^−^.

As listed in Table 1, the compounds were determined according to their characteristic MS fragmentation patterns, or the retention times of reference standards. Take an alkaloid (**17**) and a flavonoid (**56**) as examples to illustrate the resolution process of the compounds. Compound **17** is a benzylisoquinoline alkaloid (t_R_ = 2.10 min, C_19_H_23_NO_3_) yielded [M + H]^+^ ion at *m/z* 314.1754, and produced fragment ion at *m/z* 283.0286 due to parent ion peak losing CH_3_NH_2_. *m/z* 206.1181 and *m/z* 107.0875 are fragment ion peaks formed by benzyl cleavage. By comparison with literature information, this component was identified as armepavine [27]. The mass spectrogram is shown in Figure 4a. Compound **56** is an oxygen glycosides flavonoid (t_R_ = 5.70 min, C_21_H_20_O_12_) yielded [M − H]^−^ ion at *m/z* 463.0343, and produced fragment ion at *m/z* 300.9948 due to parent ion peak losing glucose moiety. By comparison with literature and reference standard information, this component was identified as isoquercetin [28]. The mass spectrogram is shown in Figure 4b.

### 2.2. Metabolomics Analysis of Six Different Parts of Lotus

Metabolomics analyses of six parts of lotus included PCA and OPLS-DA. First, to separate the parts and obtain the maximum variables, PCA was used to obtain the score plots (Figure 5a) and loading plots (Figure 5b). In the score plots, the green QC points are closely gathered together to form a cluster, which indicates that the system is stable. It can be seen from the figure that the samples from the **Sem**, **Rec**, **Plu**, **Sta**, **Fol** and **RN** groups could be easily divided into six clusters, and the six parts had achieved obvious separation, indicating that the six parts could be easily distinguished. In the loading plots, 23 variables that can be distinguished among the six clusters were found.

Second, to further evaluate the differences between the six parts, one was distinguished from the others, the maximum separation of the six parts was achieved, the potential biomarkers that may lead to the differences were found, and OPLS-DA was carried out. Then, for the visualization of the OPLS-DA and convenient interpretation of the model, S-plots were created. At the same time, to screen the different components, the variable importance of the projection (VIP) was introduced. The metabolites with VIP values above 1.0 and *p*-values below 0.05 were considered as potential biomarkers [29,30,31]. Based on these two important parameters and the identification of the components from six parts (Table 1), 23 reliable known biomarkers were found to distinguish the six parts and were labeled in the S-plots (Figure 6). In addition, a heatmap (Figure 7) was drawn to systematically evaluate these biomarkers and visually display the intensity of these biomarkers. For **Sem**, there were three potential biomarkers, including flavonoids (**110**, **125**) and a quinone (**99**). For **Fol**, there were three potential biomarkers, including terpenoids (**105**, **108**) and an alkaloid (**102**). For **Plu**, there were nine potential biomarkers, including flavonoids (**29, 47, 55, 86**), alkaloids (**28, 38, 44**) an organic acid (**10**) and an organic acid ester (**85**). For **Sta**, there were four potential biomarkers, including steroids (**133, 153**), a terpenoid (**131**) and an organic acid (**158**). For **Rec**, there were three potential biomarkers, including an alkaloid (**106**), a steroid (**135**), and an amide (**126**). For **RN**, there was only one potential biomarker–a terpenoid (**130**). These robust biomarkers enabling the differentiation among **Sem, Fol, Plu, Sta, Rec** and **RN** can be used for the rapid identification of six parts of lotus.

## 3. Discussion

Herbal medicines usually play a holistic role in maintaining health through multiple targets because they contain multiple constituents. Being a traditional Chinese herb, lotus has been used to treat various diseases. In the last decades, chemoinformatics and systems pharmacology have been successfully applied in the discovery of the active component of traditional Chinese medicines and their mechanisms of action. It is well known that the process of fully understanding the ingredients of herb using traditional methods is labor intensive, difficult and time-consuming. Fortunately, the combining of UPLC-QToF-MS technology and UNIFI platform helps researchers reveal the containing compounds in herbs in an efficient way.

The present study analysed the principal components of the lotus six parts by UPLC-QToF-MS. Combined the related literatures [6,32,33] with our previous experiments, heat reflux extraction with 80% ethanol was chosen for the samples extraction. And the detecting conditions, mobile phase elution solutions of UPLC, positive and negative ion detection modes of QToF-MS, were optimized by quality control samples. MS and MS/MS data were collected simultaneously to improve the efficiency and accuracy of data collection in the MS^E^ model. The tolerance of t_R_ was ± 0.1 min. The isotopic pattern was included in the peak identification. The permutation testing was the scoring function for identification and statistical analysis parameter *p*-value need to below 0.05. As a result, 171 compounds were identified or tentatively characterized from the six parts of lotus.

It was found that flavonoids were the most common compounds: 56 of 171 were flavonoids. **Sta** was ranked first due to the 31 kinds of flavonoids being detected, followed by **Plu** (24 kinds), **Sem** (23 kinds), **Fol** (20 kinds), and **Rec** (20 kinds). By comparing the species of flavonoids distributed in the six parts, it was found that 18 of the 20 in **Rec** are consistent with those in **Sta**, and half of the species in **Sem** are the same as those in **Plu**. Luteolin is the only flavonoid detected from **RN**, and it is also available in the other five parts.

Alkaloids are also important active components in lotus. In this experiment, 22 kinds of alkaloid compounds were detected, including isoquinoline alkaloids, aporphine alkaloids and so on. **Plu** contained 15 kinds, followed by **Fol** with 11 kinds. Armepavine and nuciferine are available in all six parts. Terpenoids were the most abundant in lotus **Sta**, containing 13 species. Statistical analysis was conducted on the compounds detected from the six parts, and the structure types and the numbers of compounds in the six parts are shown in Figure 8.

The established method can be used as a standard protocol for directly discriminating between and predicting the six parts of lotus. Traditional Chinese herbal therapy can be characterized by the use of a large number of multi-herb formulae. Chinese patent medicines (CPM), which come from traditional Chinese classical prescriptions, are usually prepared by modern pharmaceutical techniques with various herbs as raw material. After processing, the unique morphological characteristics of the original herbs disappeared, and the active chemical constituents were successfully preserved. So the chemical compositions and characters of CPM have been considered a reliable index of quality control. Being used as both delicious food and empirical medicine, the formulation and preparation conditions must be improved to achieve better delivery of nutritional ingredients and increased bioactivities of the food and medicinal products. Lotus is one of the commonly-used herbal drugs. Its two parts or more appeared in one prescription for some disease treatment. For example, *Bai Dai Wan* from *Hui Zhi Tang Jing Yan Fang* (Experiential Prescriptions from Hui Zhi Clinic) is comprised of eight herbs including lotus seed, stamens and rhizome nodes. The biomarkers found in this paper can provide bases for examining the raw herbs types in the finished preparations and improving the quality of products.

Most importantly, data analyses provided useful information for the further study and usage of the plant. The results of this phytochemical profile study are far more comparable because the same procedure was used during sample analysis, including during sample treatment, the detection procedure, data processing and the interpretation of testing data. Compared with the previous studies [2,33,34,35], this was the first time that alkaloids (**8, 17, 20, 24, 63, 95**) were detected in **Sta**, and terpenoids (**97, 124, 139, 140, 148, 154, 159, 160, 170**) were detected in **Sem**. **Sta** is a more noteworthy part than the others, for it has the greatest diversity of flavonoids and terpenoids, but relatively few studies on **Sta** can be found compared to the other parts.

Being a traditional Chinese herb, lotus has been used for more than 2,000 years. However, the material basis of a number of folk applications has not been illustrated clearly. For instance, being the hemostatic agent in traditional Chinese medicine, **Rec** and **RN** have the same indications. After comparison, we found that 15 of the 19 compounds identified in **RN** were also presented in **Rec**. They were three alkaloids (**17, 95, 127**), three terpenoids (**138, 152, 154**), one flavonoid (**91**), three steroids (**147, 149,1 62**), and five organic acids and esters (**6, 117, 165, 167, 171**). They may be the material basis of hemostatic activity of **RN** and **Rec**.

Compound **41** (liensinine) and **59** (neferine) play a major role in anti-Alzheimer disease agents [36]. However, none of them was detected in the receptacle, although **Rec** was reported to have an anti-Alzheimer effect [37]. In this paper, **41** and **59** were identified in **Sem**, **Fol** and **Plu**, as well as **41** were identified in **Rec**. The result provided useful information for the research and utilization of lotus seed, leaf and receptacle. Meanwhile, the discoveries of alkaloids in **Sta** (**8, 17, 20, 24, 63, 95**) and **Rec** (**17, 41, 43, 95, 103, 106, 127**) may be useful for explaining their anti-ischemic effect.

In summary, the holistic and intuitionistic description of the chemical constituents in lotus six parts in this paper contribute new information to the phytochemical research of lotus. The results will be helpful in illustrating the chemical basis of herbs activities. The established method and identified biomarkers provide valuable data and references for quality control of the CPMs who contain lotus different parts in the prescriptions.

The results of this research may be limited by the capacity of the identification database. Not compared with an analytical standard, identifications of compounds are presumptive. Therefore, the identification relies more on standard secondary spectra database. Only a fraction of compounds were included in the HMBD database, so relatively few are detected although compounds are varied and abundant in the six parts of lotus. In addition to the difference ingredients, differences in the content of common ingredients of herbs also affect their pharmacological activity. In the future, more efforts should be devoted to research on the concentrations of biological compounds and biomarkers of the herbs.

## 4. Materials and Methods

### 4.1. Materials and Reagents

The products of **Sem**, **Fol**, **Plu**, **Sta**, **Rec** and **RN** were collected from their respective cultivation areas or purchased from herbal markets in China. A total of 48 batches (each part 8 batches) were gathered and identified by Professor Ping-Ya Li (School of Pharmaceutical Sciences, Jilin University, Changchun, China). The voucher specimens (No. 2019224-2019272) had been deposited at the Research Center of Natural Drug, School of Pharmaceutical Sciences, Jilin University, Changchun, China. A site list of the samples collected is given in Table 2.

LC-MS grade methanol and acetonitrile were purchased from Fisher Chemical Company (Geel, Belgium). Formic acid was purchased from Sigma-Aldrich Company (St. Louis, MO, USA). Leucine enkephalin was provided by Waters Technologies Corporation (Milford, MA, USA). All other chemicals were of analytical grade. Water was purified and deionized using a Millipore water purification system. LC-MS grade six reference standards, including hyperoside, liensinine, isoliensinine, neferine, kaempferol-3-*O*-rutinoside, and N-methylcoclaurine, were isolated from our own laboratory and were previously identified and confirmed by ^1^H-NMR and ^13^C-NMR. Five standard compounds, including rutin, quercetin, kaempferol, caffeic acid, and luteolin 7-glucoside, were purchased from the China National Institutes for Food and Drug Control. Six standard compounds, including isoquercitrin, isorhamnetin, catechin, epicatechin, nuciferine, and chlorogenic acid, were purchased from Sichuan Weikeqi Biological Technology Co., Ltd. (Sichuan, China). The purity of all the regents was HPLC ≥ 98%.

### 4.2. Sample Preparation and Extraction

A mixer was used to grind each sample to generate homogeneous powders. Sample powders (0.20 g) were refluxed with 6 mL 80% (*v*/*v*) aqueous ethanol and extracted twice for 30 min each. By filtrating, the filtrate evaporated to dryness by a filtrate recovery system. The residue was dissolved with methanol; then, the mixture was filtered by a 0.22 μm syringe filter and tested by a UPLC system. A quality control (QC) solution was prepared for each part by taking 10 μL from every sample solution and then mixing. Eight QC injections were performed randomly in the testing process to ensure the stability and suitability consistency of the MS analysis. The volume injected for the samples and QC solution was 5 μL for each run.

### 4.3. UPLC-QToF-MS

To enable high sensitivity, selectivity, speed and precision, QToF technology and UPLC/MS^E^ were used in this experiment. A Xevo G2-XS QToF mass spectrometer (Waters, Milford, MA, USA) connected to an UPLC system by an electrospray ionization (ESI) interface, was used for UPLC-QToF-MS^E^. An ACQUITY UPLC BEH C_18_ (100 × 2.1 mm, 1.7 μm) column was used for sample separation. The mobile phases consisted of eluent A (0.1% formic acid in water, *v*/*v*) and eluent B (0.1% formic acid in acetonitrile, v/v). A gradient elution method was used. The elution conditions were as follows: 0-2 min, 10% B; 2–26 min, 10–100% B; 26–29 min, 100% B; 29–29.1 min, 100–10% B; 29.1–30 min, 10% B. The flow rate was 0.4 mL/min. The column temperature was 30 °C. The data were collected by mass spectrometer, the MS^E^ continuum model was used for screening analysis, and the MS^E^ centroid model was used for metabonomics analysis in MarkerLynx software. When running a single LC system, a low collision energy (CE) scan was quickly swithed to a high CE scan, from 6 V to 20–40 V. The capillary voltages were 2.6 kV (ESI^+^) and 2.2 kV (ESI^−^) and the cone voltage was 40 V. The source and desolvation temperatures were 150 °C and 400 °C, respectively. The flow rates of the cone gas and desolvent gas were 50 L/h and 800 L/h respectively. Leucine enkephalin (LE, *m/z* 556.2771 (ESI^+^), 554.2615 (ESI^−^)) was injected at a rate of 10 μL/min. The data were recorded with a MassLynx V4.1 workstation.

### 4.4. Data Analysis

For the screening analysis, the MS raw data were imported into the Waters’ UNIFI 1.7.0 platform to quickly identify the chemical components. Two hundred was set as the minimum peak area for two-dimensional peak detection. For three-dimensional peak detection, the low-energy peak intensity was over 1000 counts, and the high-energy peak intensity was over 200 counts. The mass error of the compound was within ± 5 ppm, and the retention time (t_R_) was within ± 0.1 min. The negative adducts +COOH and -H and positive adducts +H and +Na were selected.

For metabonomics analysis, the original MS^E^ data were processed by using Waters MarkerLynx XS V4.1 software, and a table of the *m/z*-t_R_ pairs with the corresponding intensities of all the peaks was obtained. The same t_R_ and *m/z* values in different batches of samples were regarded as the same component. The main parameters included the following: t_R_ range, 0–30 min; minimum intensity, 5%; mass range, 100–1500 Da; mass tolerance, 0.10. Multivariate statistical analysis was performed, including PCA and OPLS-DA.

## Figures and Tables

**Figure 1 molecules-26-01855-f001:**
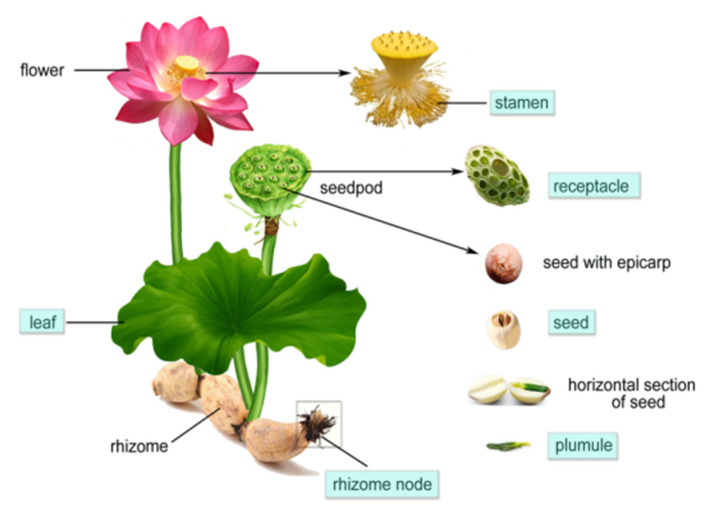
Schematic diagram of the whole lotus plant. The part listed in CP has a blue highlight in the background.

**Figure 2 molecules-26-01855-f002:**
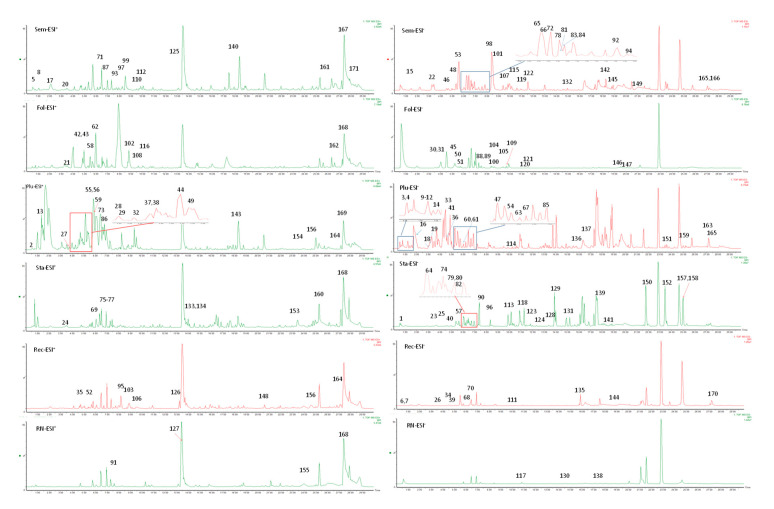
The representative BPI chromatograms of **Sem**, **Fol**, **Plu**, **Sta**, **Rec**, and **RN** in positive and negative modes.

**Figure 3 molecules-26-01855-f003:**
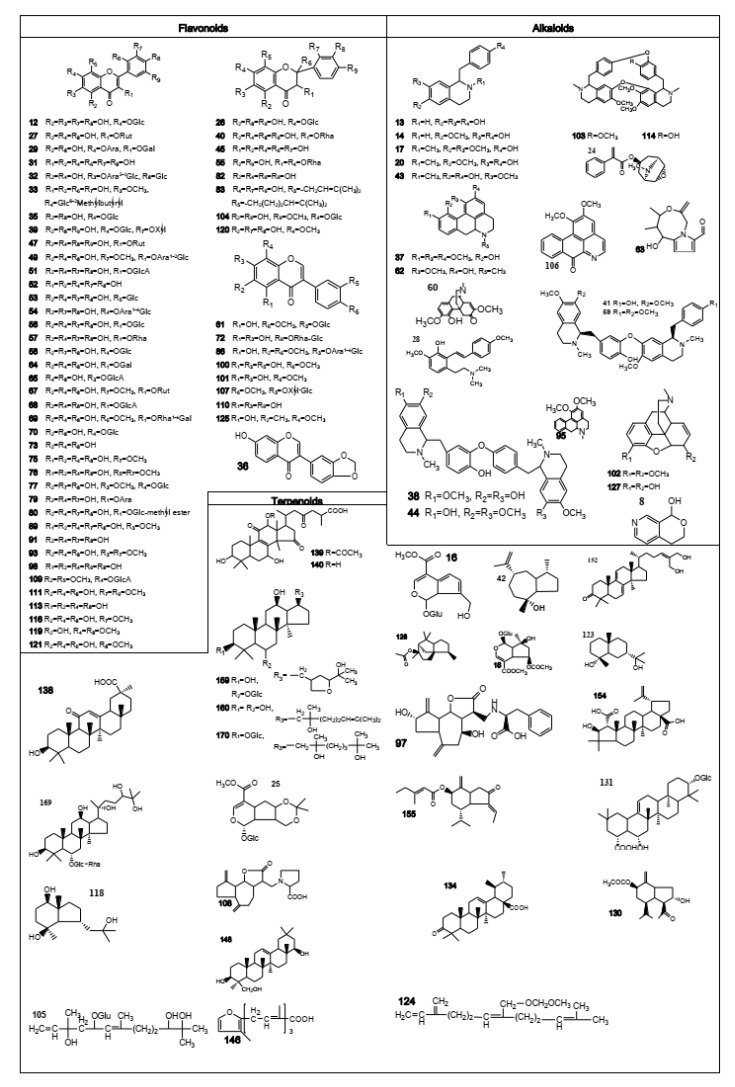
Chemical structures of compounds identified in six different parts of lotus.

**Figure 4 molecules-26-01855-f004:**
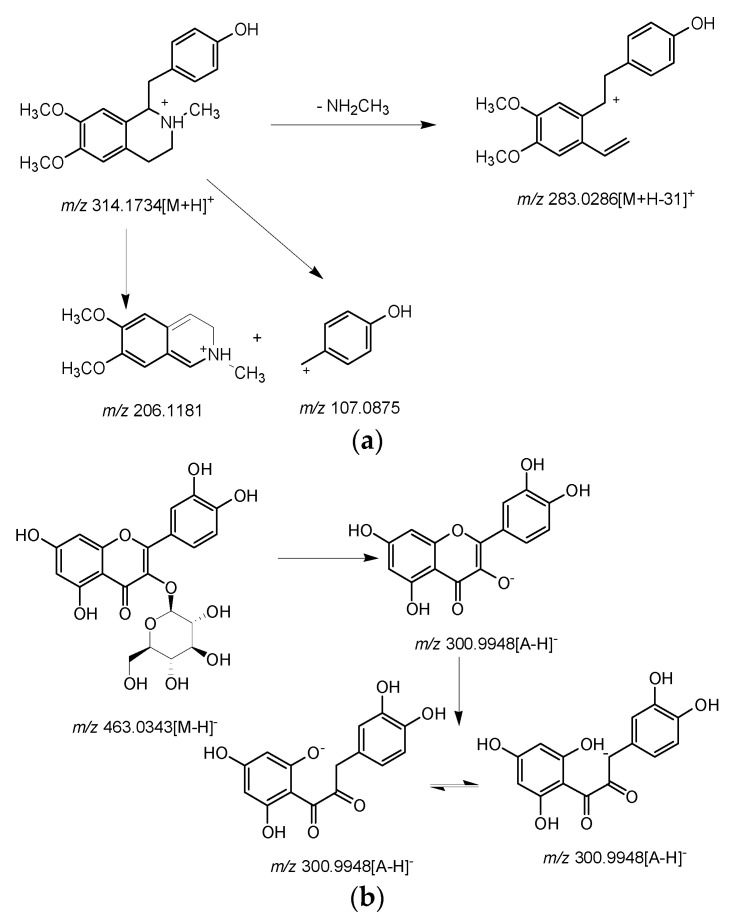
The possible fragment pathway for peak **17** and **56**.

**Figure 5 molecules-26-01855-f005:**
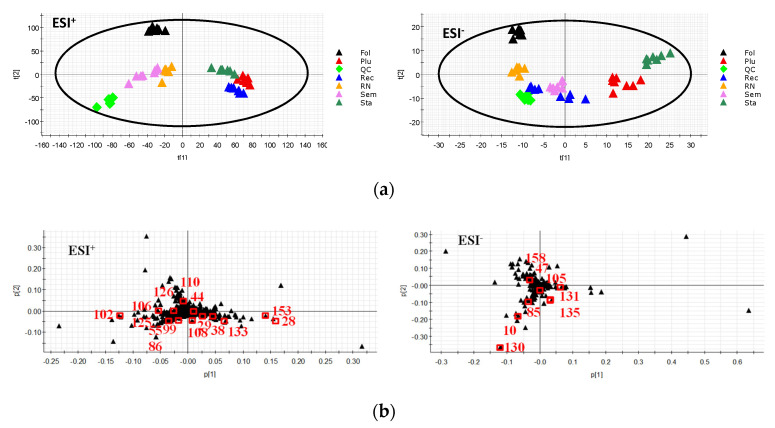
The PCA score plot (**a**) and loading plot (**b**) of six different parts of lotus.

**Figure 6 molecules-26-01855-f006:**
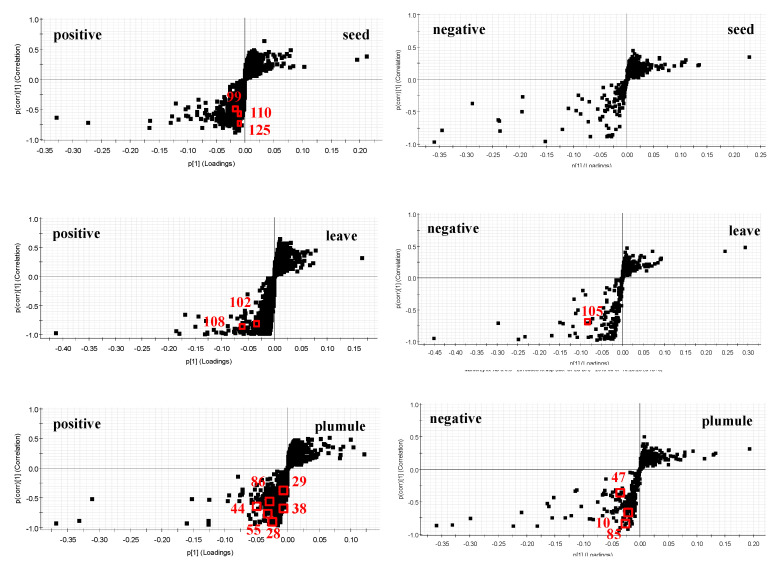
The OPLS-DA/S-plots of **Sem**, **Fol**, **Plu**, **Sta**, **Rec**, and **RN** of lotus. The points on the lower left represent the compounds in this part, and the points at the higher right represent the compounds in the other five parts. The biomarkers and their compound numbers are marked in red.

**Figure 7 molecules-26-01855-f007:**
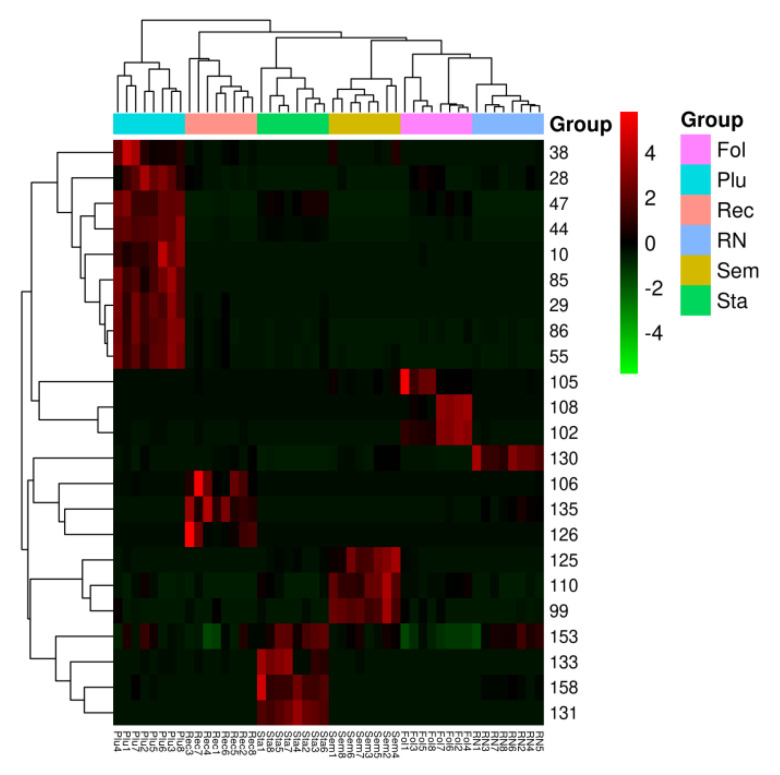
Heatmap visualizing the intensities of potential biomarkers.

**Figure 8 molecules-26-01855-f008:**
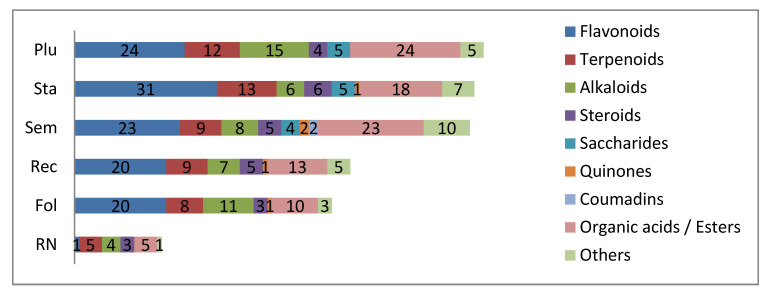
Type and number of compounds in the six parts of lotus.

**Table 1 molecules-26-01855-t001:** Compounds identified from different parts of lotus by UPLC-QToF-MS^E^.

No.	t_R_ (min)	Formula	Theoretical (Da)	Calculated Mass (Da)	Mass Error (ppm)	MS^E^ Fragmentation	Identification	Sources	Ref.
1	0.53	C_24_H_42_O_21_	666.2219	666.2226	1.12	711.1808[M + HCOO]^−^, 665.1649[M − H]^−^, 485.1080[M-H-Glc]^−^, 179.0417[Glc-H]^−^, 161.0367[Glc-H-H_2_O]^−^, 689.2103[M + Na]^+^	Nystose	Sem,Plu,Sta	^b^
2	0.55	C_18_H_32_O_16_	504.1690	504.1678	−2.40	527.1895[M + Na]^+^, 505.1786[M + H]^+^, 503.1612[M − H]^−^, 485.1170[M-H-H_2_O]^−^, 179.0417[Glc-H]^−^	Gentiotriose	Sem,Plu,Sta	^b^
3	0.57	C_5_H_10_O_5_	150.0528	150.05	0.24	195.0352[M + HCOO]^−^, 149.0345[M − H]^−^	Arabinose	Plu,Sta	^b^
4	0.57	C_12_H_22_O_11_	342.1162	342.1166	1.09	341.0829[M − H]^−^, 163.0594[M-H-Glc-H_2_O]^−^	Isomaltose	Sem,Plu,Sta	^b^
5	0.62	C_12_H_22_O_11_	342.1162	342.1152	−2.89	365.1058[M + Na]^+^, 163.0750[M+H-H_2_O]^+^	Sucrose	Sem,Plu,Sta	^b^
6	0.75	C_6_H_8_O_7_	192.0270	192.0266	−2.29	191.0192[M − H]^−^, 147.0065[M-COOH]^−^, 130.9980[M-H-COOH-CH_3_]^−^	Citric acid	Sem,Plu,Sta,Rec,RN	^b^
7	0.78	C_7_H_4_O_6_	184.0008	184.0004	−2.00	182.9704[M − H]^−^, 138.9851[M-COOH]^−^	Chelidonic acid	Plu,Rec	
8	1.09	C_9_H_11_NO_2_	165.0790	165.0785	−2.69	210.0535[M + HCOO]^−^, 164.0521[M − H]^−^	Gentiatibetine	Sem,Sta	^b^
9	1.24	C_9_H_8_O_3_	164.0473	164.0478	2.87	209.0189[M + HCOO]^−^, 163.0256[M − H]^−^, 119.0387[M-OC_2_H_5_]^−^	*p*-Coumaric acid	Plu,Rec	^b^
10	1.24	C_15_H_18_O_8_	326.1002	326.1008	1.80	325.0659[M − H]^−^, 163.0256(C_9_H_7_O_3_)	Melilotoside	Plu	^b^
11	1.24	C_16_H_20_O_10_	372.1056	372.1061	1.09	371.0664[M − H]^−^, 165.0267[M-H-Aglc-CO_2_]^−^	Deacetyl asperuloside	Plu	
12	1.24	C_21_H_20_O_12_	464.0955	464.0956	0.22	509.0485[M + HCOO]^−^, 463.0916[M − H]^−^	6-Hydroxyluteolin-7-*β*-d-glucopyranoside	Plu	[26]
13	1.45	C_16_H_17_NO_3_	271.1208	271.1205	−1.20	272.1302[M + H]^+^, 255.1011(C_16_H_15_O_3_), 237.0922(C_16_H_13_O_2_), 209.0978(C_15_H_13_O), 161.0595(C_10_H_9_O_2_), 143.0503(C_10_H_7_O), 107.0515(C_7_H_7_O)	Higenamine	Fol,Plu	^b^
14	1.52	C_17_H_19_NO_3_	285.1365	285.1370	1.66	286.1447[M + H]^+^, 284.1104[M − H]^−^, 237.0922(C_16_H_13_O_2_), 209.0949(C_15_H_13_O), 115.0560(C_9_H_7_), 107.0515(C_7_H_7_O)	Coclaurine	Sem,Plu	[26]
15	1.78	C_11_H_12_O_5_	224.0685	224.0694	4.04	223.0492[M − H]^−^, 205.0236[M-H-H_2_O]^−^, 179.0253[M-COOH]^−^	Sinapic acid	Sem,Fol,Plu	^b^
16	1.95	C_17_H_22_O_10_	386.1213	386.1208	−1.36	431.1234[M + HCOO]^−^, 385.0857[M − H]^−^	Hedyotoside	Plu	
17	2.10	C_19_H_23_NO_3_	313.1678	313.1676	−0.58	314.1734[M + H]^+^, 283.0286[M-CH_3_NH_2_]^+^, 206.1181(C_12_H_17_NO_2_), 107.0875(C_7_H_7_O)	Armepavine	Sem,Fol,Plu,Sta,Rec,RN	[26]
18	3.02	C_19_H_28_O_12_	448.1581	448.1571	−2.27	493.0695[M + HCOO]^−^, 447.1161[M − H]^−^	6-O-Acetylshanzhiside methyl ester	Plu	
19	3.08	C_18_H_26_O_10_	402.1526	402.1522	−1.00	447.0945[M + HCOO]^−^, 401.1143[M − H]^−^	Benzyl alcohol xylopyranose (1→6) glucopyranoside	Sem,Plu,Sta	^b^
20	3.31	C_18_H_21_NO_3_	299.1521	299.1517	−1.59	300.1602[M + H]^+^, 269.1184(C_17_H_17_O_3_), 237.1457(C_16_H_13_O_2_), 209.0475(C_15_H_13_O)	*N*-Methylisococlaurine	Sem,Plu,Sta	^b^
21	3.38	C_9_H_14_O_7_	234.0740	234.0728	−4.83	257.0685[M + Na]^+^, 235.0869[M + H]^+^	Trimethyl citrate	Fol	
22	3.40	C_9_H_6_O_4_	178.0266	178.0266	0.10	177.0034[M − H]^−^, 133.0152	Esculetin	Sem	
23	3.44	C_15_H_16_O_6_	292.0947	292.0946	−0.31	337.0595[M + HCOO]^−^, 291.0453[M − H]^−^	Cnidimol D	Sta	
24	3.47	C_17_H_19_NO_3_	285.1365	285.1352	−4.40	286.1412[M + H]^+^, 209.1126	Aposcopolamine	Fol,Sta	[26]
25	3.54	C_20_H_30_O_11_	446.1788	446.1790	0.52	491.1207[M + HCOO]^−^, 445.1222[M − H]^−^	Hedyoside	Sta	
26	3.69	C_21_H_22_O_11_	450.1162	450.1161	−0.24	449.0625[M − H]^−^, 431.1064[M-H-H_2_O]^−^	Miscanthoside	Sta,Rec	b
27	3.78	C_27_H_30_O_15_	594.1585	594.1580	−0.86	639.0868[M + HCOO]^−^, 593.1044[M − H]^−^	Nicotiflorin	Sem,Plu,Sta,Rec	
28	4.00	C_20_H_25_NO_3_	327.1834	327.1830	−1.36	328.1913[M + H]^+^, 300.1531,283.6312	Leonticine	Plu	[26]
29	4.03	C_26_H_28_O_15_	580.1428	580.1422	−1.14	581.1528[M + H]^+^, 547.1465	Kaempferol-3-*O*-*β*-d-glucopyranoside-7-*O*-*α*-L-arabinofuranoside	Plu	[26]
30	4.04	C_10_H_8_OS_2_	208.0017	208.0009	−3.66	252.9815[M + HCOO]^−^, 206.9634[M − H]^−^	1-(5-Thiophen-2-ylthiophen-2-yl)ethanone	Fol	
31	4.04	C_15_H_10_O_8_	318.0376	318.0385	2.96	316.9868[M − H]^−^, 298.9766[M-H-H_2_O]^−^	Myricetin	Fol	^b^
32	4.34	C_32_H_38_O_19_	726.2007	726.1977	−4.15	771.1393[M + HCOO]^−^, 725.1418[M − H]^−^	Vaccarin	Plu	
33	4.47	C_26_H_28_O_14_	564.1479	564.1487	1.33	563.1066[M − H]^−^, 493.1105	Patuletin-7-*O*-[6′′-(2-Methylbutyryl)]-glucopyranoside	Sem,Plu,Sta,Rec	[26]
34	4.54	C_8_H_8_O_4_	168.0423	168.0425	1.64	167.0135[M − H]^−^, 106.9976[M-H-CH_3_-COOH]^−^	Isovanillic acid	Fol,Rec	^b^
35	4.58	C_26_H_28_O_14_	564.1479	564.1477	−0.28	565.1638[M + H]^+^, 445.1074	Apiin	Sem,Plu,Rec	
36	4.61	C_16_H_10_O_5_	282.0528	282.0530	0.46	327.0220[M + HCOO]^−^, 281.0124[M − H]^−^, 237.0462[M-COOH]^−^	Pseudobaptigenin	Plu	^b^
37	4.65	C_19_H_21_NO_4_	327.1471	327.1466	−1.54	328.1542[M + H]^+^, 297.6281[M + H-OCH_3_]^+^, 296.1195[M + H-CH_3_OH]^+^	Norisocorydin	Fol,Plu	^b^
38	4.65	C_36_H_40_N_2_O_6_	596.2886	596.2883	−0.50	641.2319[M + HCOO]^−^, 597.2950[M + H]^+^	Dauriciline	Plu	[26]
39	4.75	C_26_H_28_O_16_	596.1377	596.1378	0.16	641.0141[M + HCOO]^−^, 595.0060[M − H]^−^	Isoetin-7-*O*-*β*-d-glucopyranosyl-2′-*O*-*β*-D-xyloypyranoside	Sta,Rec	
40	4.79	C_21_H_22_O_11_	450.1162	450.1154	−1.82	495.0541[M + HCOO]^−^, 449.0191[M-H]^−^, 431.1490[M-H-H_2_O]^−^	Astilbin	Sta	^b^
41	4.93	C_37_H_42_N_2_O_6_	610.3043	610.3034	−1.40	609.2213[M − H]^−^, 503.1425(C_30_H_35_N_2_O_5_), 489.0128(C_29_H_33_N_2_O_5_)	Liensinine	Sem,Fol,Plu,Rec	^b^
42	4.96	C_15_H_26_O	222.1984	222.1973	−4.95	267.1642[M + HCOO]^−^, 221.0231[M − H]^−^	Pogostol	Fol,Plu	^b^
43	4.96	C_18_H_21_NO_3_	299.1521	299.1517	−1.48	300.1708[M + H]^+^, 269.1218(C_17_H_17_O_3_), 237.0769(C_16_H_13_O_2_), 209.1600(C_15_H_13_O), 107.0537(C_7_H_7_O)	N-Methylcoclaurine	Fol,Rec	[26]
44	5.17	C_37_H_42_N_2_O_6_	610.3043	610.3047	0.61	611.3093[M + H]^+^, 568.2705	Dauricinoline	Plu	[26]
45	5.24	C_15_H_12_O_7_	304.0583	304.0588	1.59	349.0178[M + HCOO]^−^, 303.0161[M − H]^−^	Taxifolin	Fol,Rec	
46	5.28	C_16_H_28_O_7_	332.1835	332.1835	−0.13	377.1581[M + HCOO]^−^, 331.1492[M − H]^−^	Betulalbuside A	Sem	^b^
47	5.32	C_26_H_28_O_14_	564.1479	564.1498	3.37	609.1000[M + HCOO]^−^, 300.0008[A-H]^−^	Rutin	Plu	[26]
48	5.35	C_18_H_10_O_8_	354.0376	354.0382	1.73	355.0372[M + H]^+^, 353.0079[M − H]^−^	Mongolicumin A	Sem	
49	5.39	C_27_H_30_O_16_	610.1534	610.1525	−1.52	611.1575[M + H]^+^, 609.0798[M − H]^−^	Nelumboroside A	Sem,Plu,Sta	[26]
50	5.43	C_16_H_20_O_8_	340.1158	340.1156	−0.56	385.0656[M + HCOO]^−^, 177.0334(C_10_H_9_O_3_)	Linocinnamarin	Sem,Fol	^b^
51	5.57	C_21_H_18_O_13_	478.0747	478.0758	2.14	477.0278[M − H]^−^, 301.0090[A]^−^, 300.0008[A-H]^−^	Quercetin 3-*O*-glucuronide	Fol,Plu,Sta,Rec	^b^
52	5.64	C_15_H_10_O_7_	302.0427	302.0418	−2.73	303.0519[M + H]^+^, 237.0421	Quercetin	Fol,Plu,Sta,Rec	^b^
53	5.67	C_21_H_20_O_11_	448.1006	448.1004	−0.30	447.0599[M − H]^−^, 429.1443(C_21_H_17_O_10_), 357.0284[M-H-90]^−^, 327.0294[M-H-120]^−^, 297.0076[M-H-150]^−^	Orientin	Sem,Plu,Fol	^b^
54	5.67	C_26_H_28_O_15_	580.1428	580.1439	1.89	625.1139[M + HCOO]^−^, 579.0697[M − H]^−^	Lutelin-7-*O*-[*β*-d-apiofuranosyl(1→6)]*β*-d-glucopyranoside	Sem,Plu,Sta	[26]
55	5.67	C_27_H_30_O_14_	578.1636	578.1644	1.49	577.1150[M − H]^−^, 431.0639, 413.0504	Kaempferitrin	Plu	^b^
56	5.70	C_21_H_20_O_12_	464.0955	464.0952	−0.54	463.0343[M − H]^−^, 300.9948[A-H]^−^, 299.9901[A-2H]^−^	Isoquercetin	Sem,Plu,Sta,Rec	^a^
57	5.71	C_21_H_20_O_11_	448.1006	448.1008	0.50	493.0378[M + HCOO]^−^, 447.0382[M − H]^−^, 429.1273[M-H-H_2_O]^−^, 300.9948[M-H-Glc]^−^	Quercitrin	Fol,Sta	^b^
58	5.74	C_21_H_20_O_11_	448.1006	448.0992	−3.14	449.1170[M + H]^+^, 287.0624(C_15_H_11_O_6_)	Luteolin-7-*O*-glucoside	Fol,Plu,Sta	^b^
59	5.86	C_38_H_44_N_2_O_6_	624.3199	624.3201	0.30	625.3275[M + H]^+^, 594.2842[M+H-NH_2_CH_3_]^+^, 582.2826[M+H-CH_2_=N-CH_3_]^+^, 489.2331(C_29_H_33_N_2_O_5_), 206.1181(C_12_H_16_NO_2_), 121.0652(C_8_H_9_O)	Neferine	Sem,Fol,Plu	^b^
60	5.89	C_19_H_23_NO_4_	329.1627	329.1621	−1.90	330.1620[M + H]^+^, 328.1265[M − H]^−^	Sinomenine	Sem,Plu	
61	5.92	C_22_H_22_O_10_	446.1213	446.1201	−2.57	491.0754[M + HCOO]^−^, 427.1105[M-H-H_2_O]^−^	Sissotrin	Sem,Plu	^b^
62	5.98	C_18_H_19_NO_2_	281.1416	281.1418	0.83	282.1552[M + H]^+^, 253.1186(C_17_H_17_O_2_), 251.1113(C_17_H_15_O_2_)	Floribundine	Fol,RN	^b^
63	5.99	C_12_H_15_NO_4_	237.1001	237.0994	−3.16	282.0810[M + HCOO]^−^, 236.0730[M − H]^−^	Desmodimine	Plu,Sta	
64	6.03	C_21_H_20_O_11_	448.1006	448.1013	1.59	447.0425[M − H]^−^, 285.0017[M-H-Glc]^−^	Trifolin	Sta	^b^
65	6.17	C_22_H_20_O_10_	444.1056	444.1068	2.66	489.0943[M + HCOO]^−^, 443.0680[M − H]^−^	Apigenin-7-*O*-glucuronide	Sem	
66	6.24	C_20_H_22_O_6_	358.1416	358.1428	3.30	357.1096[M − H]^−^, 339.0475[M-H-H_2_O]^−^	Glicophenone	Sem	^b^
67	6.24	C_28_H_32_O_16_	624.1690	624.1705	2.29	669.0948[M + HCOO]^−^, 623.1085[M − H]^−^, 315.0210[A]^−^	Isorhamnetin 3-*O*-rutinoside	Plu,Sta,Rec	^b^
68	6.28	C_21_H_18_O_12_	462.0798	462.0814	3.40	507.0176[M + HCOO]^−^, 461.0183[M − H]^−^, 285.0052(C_15_H_9_O_6_)	kaempferol-3-*O*-glucuronide	Fol,Sta,Rec	^b^
69	6.29	C_28_H_32_O_16_	624.1690	624.1673	−2.81	625.1740[M + H]^+^, 607.2749[M+H-H_2_O]^+^, 317.0661(C_16_H_13_O_7_)	Isorhamnetin 3-*O*-robinobioside	Plu,Sta	^b^
70	6.34	C_21_H_20_O_10_	432.1056	432.1066	2.27	477.0591[M + HCOO]^−^, 431.0469[M − H]^−^	Cosmosiin	Sem,Plu,Sta,Rec	^b^
71	6.36	C_25_H_33_N_5_O_7_	515.2380	515.2365	−2.91	538.2259[M + Na]^+^, 516.1849[M + H]^+^	Asterinin D	Sem	
72	6.45	C_27_H_30_O_14_	578.1636	578.1632	−0.62	623.2516[M + HCOO]^−^, 577.1199[M − H]^−^	Sophorabioside	Sem,Sta	^b^
73	6.48	C_15_H_10_O_5_	270.0528	270.0522	−2.44	269.0117[M − H]^−^, 271.0582[M + H]^+^	Apigenin	Sem,Plu,Sta,Rec	^b^
74	6.52	C_43_H_42_O_22_	910.2168	910.2125	−4.72	955.0959[M + HCOO]^−^, 909.0781[M − H]^−^	Carthamin	Sta	^b^
75	6.57	C_16_H_12_O_7_	316.0583	316.0574	−2.88	361.0094[M + HCOO]^−^, 317.0589[M + H]^+^, 315.0173[M − H]^−^, 151.0941(C_7_H_3_O_4_)	Isorhamnetin	Fol,Sta,Rec	^b^
76	6.57	C_17_H_14_O_8_	346.0689	346.0677	−3.49	347.0693[M + H]^+^, 332.0417[M+H-CH_3_]^+^	Limocitrin	Sta	^b^
77	6.57	C_22_H_22_O_12_	478.1111	478.1098	−2.81	479.1059[M + H]^+^, 477.0547[M − H]^−^, 459.0726[M-H-H_2_O]^−^, 315.0173[M-Glc]^−^	Nepitrin	Sta,Rec	^b^
78	6.66	C_19_H_30_O_7_	370.1992	370.1996	1.14	415.1674[M + HCOO]^−^, 371.2011[M + H]^+^	(6R,9R)-3-Oxo-α-ionol *β*-d-glucoside	Sem	
79	6.66	C_20_H_18_O_10_	418.0900	418.0901	0.33	417.0346[M − H]^−^, 285.0052	Juglalin	Sta	^b^
80	6.66	C_22_H_20_O_13_	492.0904	492.0917	2.68	493.0865[M + H]^+^, 491.0209[M − H]^−^	Quercetin-3-*O*-*β*-d-glucuronide-6″-methyl ester	Fol,Sta,Rec	
81	6.70	C_9_H_16_O_4_	188.1049	188.1045	−1.70	187.0818[M − H]^−^, 169.0681[M-H-H_2_O]^−^, 143.0950[M-COOH]^−^	Azelaic acid	Sem,Plu,Sta	^b^
82	6.74	C_15_H_12_O_6_	288.0634	288.0641	2.31	287.0242[M − H]^−^, 125.0811(C_6_H_5_O_3_)	2-Hydroxynaringenin	Sta	^b^
83	6.74	C_25_H_28_O_5_	408.1937	408.1921	-3.85	453.1553[M + HCOO]^−^, 407.1535[M − H]^−^	2′,4′,7-Trihydroxy-6,8-bis(3-methyl-2-butenyl)flavanone	Sem	
84	6.74	C_19_H_32_O_7_	372.2148	372.2158	2.56	417.1852[M + HCOO]^−^, 371.1847[M − H]^−^, 209.0633[M-Glc]^−^, 373.2178[M + H]^+^	Blumenol C glucoside	Sem,Fol,Plu	^b^
85	6.77	C_28_H_31_ClO_10_	562.1606	562.1630	4.33	607.1154[M + HCOO]^−^, 561.1597[M − H]^−^	Physalin H	Plu	^b^
86	6.79	C_28_H_32_O_15_	608.1741	608.1741	−0.02	609.1818[M + H]^+^, 315.1604	Kakkalide	Plu	
87	6.98	C_22_H_22_O_10_	446.1213	446.1199	−3.16	469.1024[M + Na]^+^, 267.0620[M+H-Glc]^+^	Glucoobtusifolin	Sem	^b^
88	7.02	C_15_H_8_O_7_	300.0270	300.0272	0.59	298.9802[M − H]^−^, 254.9951[M-H-CO_2_]^−^	Pseudopurpurin	Fol,Sta,Rec	
89	7.02	C_16_H_12_O_8_	332.0532	332.0538	1.89	331.0454[M − H]^−^, 312.9961[M-H-H_2_O]^−^	Patuletin	Fol,Sta,Rec	^b^
90	7.41	C_28_H_34_O_9_	514.2203	514.2185	−3.44	513.1503[M − H]^−^, 471.0800[M-COCH_3_]^−^	Nomilin	Sta	^b^
91	7.53	C_15_H_10_O_6_	286.0477	286.0475	−0.99	287.0624[M + H]^+^, 285.0155[M − H]^−^	Luteolin	Sem,Fol,Plu,Sta,Rec,RN	^b^
92	7.84	C_19_H_12_O_7_	352.0583	352.0579	−1.19	397.0275[M + HCOO]^−^, 351.0215[M − H]^−^	Phellibaumin A	Sem,Rec	
93	7.96	C_17_H_14_O_7_	330.0740	330.0754	4.36	353.0627[M + Na]^+^, 331.2370[M + H]^+^	Jaceosidin	Sem	
94	8.08	C_25_H_24_O_12_	516.1268	516.1254	−2.65	515.0680[M − H]^−^, 352.9925(C_16_H_17_O_9_)	1,3-Dicaffeoylquinic acid	Sem	^b^
95	8.12	C_19_H_21_NO_2_	295.1572	295.1568	−1.35	296.1688[M + H]^+^, 265.1266(C_18_H_17_O_2_), 250.0978(C_17_H_14_O_2_), 235.0775(C_16_H_11_O_2_), 219.0819(C_16_H_11_O), 191.0874(C_15_H_11_), 179.0890(C_14_H_11_)	Nuciferine	Sem,Fol,Plu,Sta,Rec,RN	[26]
96	8.36	C_11_H_16_O_3_	196.1099	196.1107	3.80	241.0896[M + HCOO]^−^, 197.1146[M + H]^+^	Loliolide	Sem,Fol,Plu,Sta	
97	8.43	C_24_H_29_NO_6_	427.1995	427.1996	0.25	450.1884[M + Na]^+^, 426.1630[M − H]^−^	Pulchellamine D	Sem,Plu	
98	8.50	C_15_H_10_O_7_	302.0427	302.0433	2.17	301.0054[M − H]^−^, 245.1061(C_13_H_9_O_5_)	Morin	Sem,Fol,Sta,Rec	^b^
99	8.53	C_15_H_10_O_6_	286.0477	286.0469	−3.10	287.0555[M + H]^+^, 269.0413[M+H-H_2_O]^+^	Citreorosein	Sem	^b^
100	8.65	C_16_H_12_O_6_	300.0634	300.0622	−3.94	345.0161[M + HCOO]^−^, 177.0334[M-C_6_H_4_OCH_3_]^−^	Pratensein	Fol	^b^
101	8.75	C_16_H_12_O_5_	284.0685	284.0682	−0.95	329.0322[M + HCOO]^−^, 282.9966[M − H]^−^	Biochanin A	Sem	^b^
102	8.84	C_19_H_21_NO_3_	311.1521	311.1527	1.93	312.1680[M + H]^+^, 254.1317	Thebaine	Fol	[26]
103	8.89	C_38_H_42_N_2_O_6_	622.3043	622.3031	−1.87	623.3239[M + H]^+^, 580.9211	Tetrandrine	Plu,Rec	[26]
104	9.32	C_22_H_24_O_11_	464.1319	464.1319	0.08	463.0740[M − H]^−^, 301.0338[M-Glc]^−^	Hesperetin-7-glucoside	Fol	^b^
105	9.40	C_21_H_38_O_9_	434.2516	434.2533	3.98	479.1637[M + HCOO]^−^, 433.1967[M − H]^−^	Amarantholidol A glycoside	Fol	
106	9.51	C_18_H_13_NO_3_	291.0895	291.0885	−3.47	314.0791[M + Na]^+^, 292.0993[M + H]^+^	Lysicamine	Rec	
107	9.64	C_27_H_30_O_13_	562.1686	562.1677	−1.69	561.1063[M − H]^−^, 115.9048	Kushenol O	Sem	
108	9.67	C_20_H_27_NO_4_	345.1940	345.1946	1.78	368.1962[M + Na]^+^, 346.2238[M + H]^+^	Saussureamine B	Fol	
109	9.68	C_22_H_20_O_11_	460.1006	460.1007	0.32	505.0414[M + HCOO]^−^, 459.0331[M − H]^−^	Wogonoside	Fol	
110	9.74	C_15_H_10_O_5_	270.0528	270.0518	−3.95	271.0582[M + H]^+^, 253.0470[M+H-H_2_O]^+^	Genistein	Sem	^b^
111	9.82	C_17_H_14_O_7_	330.0740	330.0724	-4.73	375.1564[M + HCOO]^−^, 329.0248[M − H]^−^	Tricin	Sta,Rec	
112	9.86	C_11_H_14_O_5_	226.0841	226.0852	4.64	249.0733[M + Na]^+^, 227.1603[M + H]^+^	Genipin	Sem	^b^
113	9.89	C_15_H_10_O_6_	286.0477	286.0481	1.21	285.0052[M − H]^−^, 286.0086, 243.0011, 174.9307, 106.9976	Kaempferol	Fol,Sta	^b^
114	9.89	C_37_H_40_N_2_O_6_	608.28863	608.2901	2.33	653.2299[M + HCOO]^−^, 607.1910[M − H]^−^	Berbamine	Plu	
115	9.96	C_23_H_18_O_8_	422.1002	422.0996	−1.37	467.0750[M + HCOO]^−^, 421.0505[M − H]^−^	Interfungin B	Sem	
116	10.08	C_16_H_12_O_6_	300.0634	300.0623	−3.59	301.0681[M + H]^+^, 283.1732[M+H-H_2_O]^+^, 271.0582(C_15_H_11_O_5_)	Chrysoeriol	Sem,Fol	^b^
117	10.88	C_18_H_34_O_5_	330.2406	330.2403	−0.85	329.1920[M − H]^−^, 313.1157[M-OH]^−^	Sanleng acid	Sem,Plu,Sta,Rec,RN	[26]
118	11.06	C_15_H_28_O_3_	256.2038	256.2038	−0.25	301.1689[M + HCOO]^−^, 255.7936[M − H]^−^	Bullatantriol	Sta	
119	11.13	C_17_H_14_O_5_	298.0841	298.0833	−2.64	343.0448[M + HCOO]^−^, 271.1218(C_15_H_11_O_5_), 267.0337[M-OCH_3_]^−^	5-Hydroxy-7-methoxy-2-(4-methoxyphenyl)-4H-chromen-4-one	Sem	^b^
120	11.27	C_16_H_14_O_6_	302.0790	302.0799	2.89	347.0313[M + HCOO]^−^, 301.0374[M − H]^−^, 283.9966[M-H-H_2_O]^−^, 164.9977(C_8_H_5_O_4_)	Blumeatin	Fol	^b^
121	11.52	C_16_H_12_O_6_	300.0634	300.0636	0.86	345.0199[M + HCOO]^−^, 299.0297[M − H]^−^, 285.0086[M-CH_3_]^−^, 284.0034[M-H-CH_3_]^−^	Diosmetin	Sem,Fol,Plu,Sta	^b^
122	11.70	C_15_H_8_O_5_	268.0372	268.0385	4.84	313.0034[M + HCOO]^−^, 267.0739[M − H]^−^	Coumesterol	Sem	^b^
123	11.73	C_15_H_28_O_2_	240.2089	240.2096	2.74	285.1746[M + HCOO]^−^, 239.1349[M − H]^−^	Isodonsesquitin A	Sta	
124	12.33	C_17_H_26_O_2_	262.1933	262.1939	2.44	307.1509[M + HCOO]^−^, 261.1005[M − H]^−^	(Z)-7-Acetoxy-methyl-11-methyl-3-methylenedodeca-1,6,10-triene	Sem,Plu,Sta	
125	13.08	C_16_H_12_O_5_	284.0685	284.0673	−4.07	285.0765[M + H]^+^, 253.1413[M-OCH_3_]^+^	Prunetin	Sem	^b^
126	13.24	C_18_H_23_NO_3_	301.1678	301.1693	4.98	324.1644[M + Na]^+^, 302.1018[M + H]^+^	Futoamide	Rec	^b^
127	13.58	C_17_H_19_NO_3_	285.1365	285.1370	1.91	308.1296[M + Na]^+^, 288.2559	Morphine	Fol,Rec,RN	^b^
128	13.75	C_17_H_28_O_2_	264.2089	264.2092	0.85	309.1676[M + HCOO]^−^, 263.1494[M − H]^−^, 221.1235[M-COCH_3_]^−^	Cedryl acetate	Plu,Sta	^b^
129	13.89	C_17_H_30_O_2_	266.2246	266.2248	0.85	311.1837[M + HCOO]^−^, 265.1130[M − H]^−^	Cireneol G	Sem,Plu,Sta,Rec	
130	14.57	C_17_H_26_O_4_	294.1831	294.1832	0.40	339.1569[M + HCOO]^−^, 293.1417[M − H]^−^	Gmelinin B	RN	
131	14.92	C_36_H_58_O_9_	634.4081	634.4058	−3.66	679.3038[M + HCOO]^−^, 633.3121[M − H]^−^	Ecliptasaponin D	Sta	

132	14.96	C_18_H_34_O_4_	314.2457	314.2463	1.89	313.2063[M − H]^−^, 239.1380[M-H-OC_4_H_9_]^−^	Dibutyl decanedioate	Sem,Plu,Sta	^b^
133	14.98	C_34_H_52_O_9_	604.3611	604.3606	−0.87	605.3710[M + H]^+^, 603.2665[M − H]^−^	Periplocoside M	Sta	
134	15.04	C_30_H_46_O_3_	454.3447	454.3440	−1.53	455.3550[M + H]^+^, 437.3432[M+H-H_2_O]^+^, 409.3499[M-COOH]^+^	Ursonic acid	Fol,Sta,Rec	^b^
135	15.77	C_29_H_46_O_4_	458.3396	458.3388	−1.75	503.2620[M + HCOO]^−^, 457.2710[M − H]^−^	Neotigogenin acetate	Rec	
136	15.81	C_18_H_32_O_2_	280.2402	280.2400	−0.93	325.2062[M + HCOO]^−^, 279.1994[M − H]^−^, 261.1038[M-H-H_2_O]^−^	Linoleic acid	Sem,Plu,Sta	^b^
137	16.37	C_15_H_22_O_4_	266.1518	266.1517	−0.27	265.1230[M − H]^−^, 221.8184[M-H-COO]^−^	Artemin	Sem,Plu,Sta	^b^
138	17.54	C_30_H_46_O_4_	470.3396	470.3375	−4.47	469.2683[M − H]^−^, 451.1740[M-H-H_2_O]^−^, 425.2926[M-COOH]^−^	Glycyrrhetinic acid	Rec,RN	^b^
139	17.58	C_32_H_44_O_9_	572.2985	572.2993	1.41	573.3043[M + H]^+^, 555.2916[M+H-H_2_O]^+^	Ganoderic acid H	Sem,Plu,Sta,Rec	^b^
140	17.77	C_30_H_44_O_8_	532.3036	532.3027	−1.75	555.2916[M + Na]^+^, 515.3719[M+H-H_2_O]^+^, 497.3564(C_30_H_41_O_6_)	Ganoderic acid G	Sem,Plu,Sta,Rec	^b^
141	18.25	C_27_H_47_N_3_O_8_	541.3363	541.3380	3.13	586.2817[M + HCOO]^−^, 540.2609[M − H]^−^	2(1H)-Isoquinolinecarboximidamide,3,4-dihydro-*N*-3,6,9,12,15,18,21,24-octaoxapentacos-1-yl-	Sem,Plu,Sta,Rec	
142	18.36	C_34_H_46_O_9_	598.3142	598.3119	−3.85	643.3286[M + HCOO]^−^, 597.2510[M − H]^−^	Daturametelin H	Sem,Sta	
143	18.45	C_18_H_28_O_2_	276.2089	276.2076	−4.83	321.1788[M + HCOO]^−^, 277.2154[M + H]^+^, 275.1695[M − H]^−^, 257.1711[M-H-H_2_O]^−^	Stearidonic acid	Sem,Plu,Sta	^b^
144	18.85	C_18_H_34_O_3_	298.2508	298.2512	1.26	343.2193[M + HCOO]^−^, 297.2159[M − H]^−^	Ricinoleic acid	Sem,Plu,Sta,Rec	^b^
145	18.93	C_39_H_60_O_15_	768.3932	768.3970	4.92	813.3219[M + HCOO]^−^, 767.2943[M − H]^−^	Hypoglaucin H	Sem	
146	19.56	C_20_H_28_O_3_	316.2038	316.2032	−2.08	361.1339[M + HCOO]^−^, 315.1700[M − H]^−^	Saurufuran B	Fol	
147	19.81	C_35_H_60_O_6_	576.4390	576.4394	0.68	621.3870[M + HCOO]^−^, 464.1719[M-C_8_H_16_]^−^	Daucosterol	Fol,Plu,Sta,Rec,RN	[26]
148	20.56	C_30_H_50_O_3_	458.3760	458.3767	1.46	459.3841[M + H]^+^, 441.3736[M+H-H_2_O]^+^, 423.3613(C_30_H_47_O)	Soyasapogenol B	Sem,Rec	^b^
149	21.23	C_28_H_46_O	398.3549	398.3550	0.22	421.3392[M + Na]^+^, 399.3546[M + H]^+^	24-Methylenecholesterol	Sem,Fol,Plu,Sta,Rec,RN	^b^
150	21.76	C_18_H_30_O_2_	278.2246	278.2252	2.27	323.1931[M + HCOO]^−^, 277.1808[M − H]^−^, 233.2307[M-COOH]^−^	Linolenic acid	Sem,Fol,Plu,Sta,Rec	^b^
151	23.35	C_19_H_38_O_2_	298.2872	298.2876	1.52	343.2383[M + HCOO]^−^, 279.1925[M-H-H_2_O]^−^, 253.1802[M-COOH]^−^	Nonadecanoic Acid	Plu,Rec	^b^
152	23.46	C_30_H_46_O_3_	454.3447	454.3437	−2.29	453.2730[M − H]^−^, 423.2258(C_29_H_43_O_2_)	Ganoderiol F	Sta,Rec,RN	^b^
153	23.47	C_29_H_48_O_2_	428.3654	428.3638	−3.76	429.3717[M + H]^+^, 411.3596[M+H-H_2_O]^+^	3*β*-Hydroxystigmast-5-en-7-one	Sta	^b^
154	23.97	C_30_H_46_O_5_	486.3345	486.3337	−1.78	509.3112[M + Na]^+^, 469.3638[M+H-H_2_O]^+^, 423.3571[M-COOH-H_2_O]^+^	Ceanothic acid	Sem,Fol,Plu,Sta,Rec,RN	^b^
155	24.35	C_21_H_30_O_3_	330.2195	330.2195	0.12	353.2165[M + Na]^+^, 331.3078[M + H]^+^	Tussilagonone	Fol,RN	^b^
156	24.66	C_22_H_42_O_2_	338.3185	338.3173	−3.38	383.2634[M + HCOO]^−^, 339.3274[M + H]^+^, 321.3164[M+H-H_2_O]^+^, 303.3048(C_22_H_39_)	Erucic acid	Sem,Fol,Plu,Sta,Rec	^b^
157	24.99	C_16_H_32_O_2_	256.2402	256.2410	3.06	255.2143[M − H]^−^, 237.1881[M-H-H_2_O]^−^, 211.1025[M-COOH]^−^	Palmitic acid	Sem,Plu,Sta	^b^
158	24.99	C_20_H_40_O_2_	312.3028	312.3028	−0.15	311.2523[M − H]^−^, 293.1382[M-H-H_2_O]^−^	Arachidic acid	Sta	^b^
159	24.99	C_36_H_62_O_10_	654.4343	654.4335	−1.15	699.3741[M + HCOO]^−^, 653.3450[M − H]^−^	Pseudo-ginsenoside RT4	Sem,Plu,Rec	
160	25.55	C_30_H_52_O_4_	476.3866	476.3865	−0.06	499.3885[M + Na]^+^, 477.3926[M + H]^+^	20(S)-Protopanaxatriol	Sem,Fol,Plu,Sta	
161	25.68	C_22_H_41_NO	335.3188	335.3178	−3.05	358.3636[M + Na]^+^, 336.3233[M + H]^+^	*N*-Isobutyl-2E,4E-octadecadienamide	Sem,Sta,Rec	
162	26.57	C_29_H_46_O	410.3549	410.3538	−2.58	411.3720[M + H]^+^, 393.3600[M+H-H_2_O]^+^	Corbisterol	Sem,Fol,Plu,Sta,Rec,RN	^b^
163	27.18	C_57_H_98_O_6_	878.7363	878.7374	1.18	923.6272[M + HCOO]^−^, 877.4003[M − H]^−^	Linolein	Sem,Plu	
164	27.19	C_29_H_46_O_2_	426.3498	426.3483	−3.45	427.3565[M + H]^+^, 409.3457[M+H-H_2_O]^+^	Stigmast-4-ene-3,6-dione	Sem,Plu,Rec	^b^
165	27.36	C_18_H_36_O_2_	284.2715	284.2641	1.51	283.2275[M − H]^−^, 255.1946[M-C_2_H_5_]^−^, 237.0084[M-H-C_2_H_5_OH]^−^	Ethyl hexadecanoate	Sem,Fol,Plu,Sta,Rec,RN	^b^
166	27.36	C_21_H_42_O_2_	326.3185	326.3195	3.12	371.2715[M + HCOO]^−^, 325.1361[M − H]^−^, 307.1581[M-H-H_2_O]^−^	Heneicosanoic acid	Plu,Sta,Rec	^b^
167	27.38	C16H_22_O4	278.1518	278.1512	−2.16	301.1426[M + Na]^+^, 279.1635[M + H]^+^, 205.2028[M-OC_4_H_9_]^+^, 57.0751(C_4_H_9_)	Diisobutyl phthalate	Sem,Fol,Plu,Sta,Rec,RN	^b^
168	27.38	C_24_H_38_O_4_	390.2770	390.2764	−1.56	413.2744[M + Na]^+^, 391.2936[M + H]^+^	Bis (2-ethylhexyl) phthalate	Sem,Fol,Plu,Sta,RN	^b^
169	27.38	C_42_H_74_O_15_	818.5028	818.5049	2.61	863.3998[M + HCOO]^−^, 819.5060[M + H]^+^	Quinquenoside L9	Plu,Sta	
170	27.4	C_36_H_64_O_9_	640.4550	640.4522	−4.50	685.3537[M + HCOO]^−^, 639.3768[M − H]^−^	3-*O*-*β*-d-Glucopyranosyl-dammar-3*β*,12*β*,20R,25-tetraol	Sem,Rec	[26]
171	27.57	C_55_H_74_N_4_O_5_	870.5659	870.5654	−0.63	893.7276[M + Na]^+^, 871.5701[M + H]^+^	Pheophytin a	Sem,Fol,Plu,Sta,Rec,RN	

^a^ Identified with the standard. ^b^ In comparison to spectral data obtained from the Human Metabolome Database (Canada).

**Table 2 molecules-26-01855-t002:** The collection sites of the tested samples.

Species	Hunan	Shandong	Hubei	Hebei	Anhui	Fujian	Jiangxi	Guangdong
Sem	Sem_1_	Sem_2_	Sem_3_	Sem_4_	Sem_5_	Sem_6_	Sem_7_	Sem_8_
Fol	Fol_1_	Fol_2_	Fol_3_	Fol_4_	Fol_5_	Fol_6_	Fol_7_	Fol_8_
Plu	Plu_1_	Plu_2_	Plu_3_	Plu_4_	Plu_5_	Plu_6_	Plu_7_	Plu_8_
Sta	Sta_1_	Sta_2_	Sta_3_	Sta_4_	Sta_5_	Sta_6_	Sta_7_	Sta_8_
Rec	Rec_1_	Rec_2_	Rec_3_	Rec_4_	Rec_5_	Rec_6_	Rec_7_	Rec_8_
RN	RN_1_	RN_2_	RN_3_	RN_4_	RN_5_	RN_6_	RN_7_	RN_8_

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
