# Peer review of "Comparative Analysis of Chemical Constituents in Different Parts of Lotus by UPLC and QToF-MS"

_molecules, 2021, doi:10.3390/molecules26071855_

Round 1
Reviewer 1 Report
The study presents a comparative analysis of different parts of lotus by UPLC and QToF-MS. The manuscript present data of real pharmacological interest it fit in the new trend of plants chemical characterisation by HRMS.
However, I feel that the ‘Discussion’ section should be improved. Given the complexity of the data processing process, many details regarding the selection of the reliable data provided the UNIFI platform should be mentioned. Many details regarding the analytical part: the optimisation of the extraction, UHPLC methods and MS methods, could be useful also.
In the ‘discussion’ part, poor connection to the aims declared in the introduction is observed. Please, improve the paragraph between 202-209 lines in such a way that it appears that the study has fulfilled its objectives. Which are the main compounds which emerge from the PCA analysis responsible for presumed biological actions? Which of the plant parts is richer in bioactive compounds? Add some conclusion from PCA analysis. Also in the section between 172-184 lines, introduce several compounds into the discussion not only luteolin, armepavine and nuciferine.
Add comparative data with other studies regarding the identification of the active compounds.
In addition it should be mentioned that these identifications are presumptive. A ‘real’ identification involves comparative analysis with an analytical standard, especially for the bioactive compounds whose structure has many similarities.
Were quantitative analyses also made? R94: ‘By comparing the quantities of the detected compounds’. In the ‘Material and method’ section a quantitative method is not described.

Author Response
Q1. The ‘Discussion’ section should be improved. Given the complexity of the data processing process, many details regarding the selection of the reliable data provided the UNIFI platform should be mentioned. Many details regarding the analytical part: the optimisation of the extraction, UHPLC methods and MS methods, could be useful also. Answer: Some details of the identification process were added in L186-190 in the revised manuscript. Details on the analysis part were added in L181-185. Q2.In the ‘discussion’ part, poor connection to the aims declared in the introduction is observed. Please, improve the paragraph between 202-209 lines in such a way that it appears that the study has fulfilled its objectives. Which are the main compounds which emerge from the PCA analysis responsible for presumed biological actions? Which of the plant parts is richer in bioactive compounds? Add some conclusion from PCA analysis. Also in the section between 172-184 lines, introduce several compounds into the discussion not only luteolin, armepavine and nuciferine. Answer: The aims were supplemented and revised in the ‘discussion’ part. Please see L249-253. The main compounds in plant parts were analyzed and discussed in L236-241 and L244-248. Q3. Add comparative data with other studies regarding the identification of the active compounds. Answer: Comparison with other studies on the identification of active compounds were supplemented in L229-231 and L236-241. Q4. In addition it should be mentioned that these identifications are presumptive. A ‘real’ identification involves comparative analysis with an analytical standard, especially for the bioactive compounds whose structure has many similarities. Answer: The relevant content was supplemented in L254-262. Q5. Were quantitative analyses also made? R94: ‘By comparing the quantities of the detected compounds’. In the ‘Material and method’ section a quantitative method is not described. Answer: The sentence of ‘By comparing the quantities of the detected compounds... ’ had been modified ‘By comparing the numbers of the detected compounds...'Reviewer 2 Report
The manuscript is generally well written, interesting and the research is properly conducted. Some minor corrections should be made, as follows:
Page 8 line 247 -248 – Rather should be „0.1% (v/v) formic acid in water”, and „.. 0,1% (v/v) formic acid in acetonitrile…”. Please complete the information on which of the two solutions is phase A and which is phase B.
Page 8 line 270 – should be „…tR ..” instead of „…tR ..”
Author Response
Q1.Page 8 line 247 -248 – Rather should be „0.1% (v/v) formic acid in water”, and „.. 0,1% (v/v) formic acid in acetonitrile…”. Please complete the information on which of the two solutions is phase A and which is phase B.
Answer: Mobile phase information had been modified as the suggestion.
Q2. Page 8 line 270 – should be „…tR ..” instead of „…tR ..”
Answer: “tR” had been modified as “tR”.
Round 2
Reviewer 1 Report
The reviewer comments were addressed. The paper should be published in the present form.